# The Roles of Autoimmunity and Biotoxicosis in Sick Building Syndrome as a “Starting Point” for Irreversible Dampness and Mold Hypersensitivity Syndrome

**DOI:** 10.3390/antib9020026

**Published:** 2020-06-22

**Authors:** Tamara Tuuminen

**Affiliations:** Kruunuhaka Medical Center, Kaisaniemenkatu 1Ba, 00 180 Helsinki, Finland; tuuminen@gmail.com

**Keywords:** neurological symptoms, mycotoxins, biotoxicosis, sensory receptors, dampness and mold, hypersensitivity syndrome, sick building syndrome

## Abstract

**Background**: The terminology of “sick building syndrome” (SBS), meaning that a person may feel sick in a certain building, but when leaving the building, the symptoms will reverse, is imprecise. Many different environmental hazards may cause the feeling of sickness, such as high indoor air velocity, elevated noise, low or high humidity, vapors or dust. **The Aim**: To describe SBS in connection with exposure to indoor air dampness microbiota (DM). **Methods**: A search through Medline/Pubmed. **Results and Conclusions**: Chronic course of SBS may be avoided. By contrast, persistent or cumulative exposure to DM may make SBS potentially life-threatening and lead to irreversible dampness and mold hypersensitivity syndrome (DMHS). The corner feature of DMHS is acquired by dysregulation of the immune system in the direction of hypersensitivities (types I–IV) and simultaneous deprivation of immunity that manifests as increased susceptibility to infections. DMHS is a systemic low-grade inflammation and a biotoxicosis. There is already some evidence that DMHS may be linked to autoimmunity. Autoantibodies towards, e.g., myelin basic protein, myelin-associated glycoprotein, ganglioside GM1, smooth muscle cells and antinuclear autoantibodies were reported in mold-related illness. DMHS is also a mitochondropathy and endocrinopathy. The association of autoimmunity with DMHS should be confirmed through cohort studies preferably using chip-based technology.

## Highlights

Sick Building Syndrome (SBS) is an imprecise terminology that may comprise a variety of environmental hazards;The term SBS does not reflect the effect of indoor air dampness microbiota that may have a devastating pathophysiological impact on human health;When left untreated, reversible SBS may proceed into irreversible dampness and mold hypersensitivity syndrome (DMHS) that may affect many organs;Autoimmunity may be linked to postural tachycardia syndrome (POTS) and myalgic encephalomyelitis/chronic fatigue syndrome (ME/CFS), the conditions associated with DMHS;Chip-based affordable technology to detect autoantibodies in patients with symptoms of peripheral nervous system is of a big clinical demand;New techniques to study indoor air toxicity should be implemented;Psychologization of SBS/DMHS should be discouraged because this approach undermines patients’ trust towards the medical community.

## 1. What is a Sick Building Syndrome?

The colorful term of sick building syndrome (SBS) was coined already in the 1970s. Originally, it was linked to the working environment [1,2,3,4,5] but later also to homes, with sick house syndrome [6]. The prominent feature of SBS is that a person experiences a set of transient symptoms in a building that makes him/her feel “sick”. The symptoms become relieved after exiting the building. SBS was also called “indoor climate syndrome” and “sick hospital syndrome” but the synonym “tight building syndrome” better suits the context of the problem; however, this term is less often used [7]. Importantly, tight buildings do not “breath” and therefore create a milieu for the growing of dampness microbiota (DM), especially in situations when construction encounters any water leakage. The core symptoms such as lethargy, mucous membrane, skin and eye irritation, headache, sore throat, non-productive cough, fatigue and cognitive problems have been described. Prolonged stays in a “sick building” may lead to a chronic cause, even up to potentially life-threatening hypersensitivity pneumonitis [8,9] or allergic alveolitis [10]. In the 1980s, it was postulated that “the deadliest pollutants of all may be the ones you breathe at home or at work” [11]. WHO has then estimated that up to 30% of new or renovated office buildings are the so-called “sick buildings” and that between 10 and 30% of the occupants may feel affected [5]. Later, in parallel to the studies of environmental toxicology in SBS and probably due to the increasing number of litigation problems worldwide and obvious conflict of interests, this disease has been labeled as “a functional somatic syndrome” or “somatization” or “idiopathic building intolerance”. It has been also labeled as a fashionable diagnosis. SBS patients were blamed for having illness as a way of life [12,13]. The major problem of the SBS terminology is its imprecise message leading to misinterpretation and even denial. Indeed, an occupant may feel unhealthy due to higher concentration of volatile organic substances, (e.g., formaldehyde or 2-ethyl-1-hexanol), nitric dioxide, ozone or due to mycotoxins expelled by growing dampness microbiota. Further, indoor air discomfort such as too high air velocity, too high temperature, insufficient ventilation or noise may make a building “sick”, but notably, these factors are not attributed to disease chronicity. Here, I will discuss SBS only in conjunction with the effects of DM.

## 2. How Sick Building Syndrome is Related to the Dampness and Mold Hypersensitivity Syndrome (DMHS)?

The impact of DM and mycotoxins on the initiation of SBS and further deterioration of many bodily functions has become already established [14,15,16,17,18] even if water damage has been removed [14]. Infestation of a building with DM as a cause of SBS deserves special mention and will be referred here in the context with SBS that constitutes the third criterion of a larger syndrome, the so-called dampness and mold hypersensitivity syndrome (DMHS) [19]. DMHS is caused by poor indoor air polluted by decay products of construction materials and organic substances emitted by growing microbiota in a water-damaged building. The clinical features of DMHS were laid in 2017, as: (a) the history of water damage in a building; (b) increased morbidity due to infections occurring in the early stage of the disease; (c) sick building syndrome, the symptoms reported by occupants that occur at the early stages of the disease; (d) the development of multiple chemical sensitivity (MCS); and (e) increased scent sensitivity such as the ability to smell a moldy odor, e.g., from the clothes of a person standing nearby. Importantly, (d) and (e) are observed in a chronic phase of DMHS, and at this stage, the disease is often irreversible, although at the avoidance of the inflicting agents, the patient can cope with the challenges of everyday life. The involvement of the central nervous system (CNS) into the pathology of SBS (as a feature of DMHS) was reported already in the 1990s. It may manifest as a concurrent chronic fatigue syndrome (CFS) [20] or MCS [9,21]. A schematic pyramid of the progression of the clinical manifestations of DMSH is shown in Figure 1. An occupant of a water-damaged building may experience also musculoskeletal pain, fibromyalgia, gastro-intestinal problems and symptoms of autonomous dysregulation. The underlying mechanisms for the plethora of neurological symptoms were hypothesized as neurogenic inflammation and neurogenic switching, the processes distinct from immune-mediated pathology [22,23]. Prolonged exposure to chemical and/or biological pollutants may overstimulate sensory receptors of the mucosa. This initiates the release of substance P and other neuropeptides involved in local or systemic inflammatory responses. When sensory receptors become overstimulated, the disease may manifest as MCS, reactive airway distress syndrome (RADS) or reactive upper airway distress syndrome (RUDS) [21,24]. In my opinion, in DMHS, the central wording of “hypersensitivity” has a prominent meaning and depicts not only the quite rarely observed development of IgE-class antibody-mediated allergy reactions, that are a type I hypersensitivity (according to the Gell and Coombs classification) towards cognate allergens of DM. Rather, hypersensitivity should be understood more broadly as a spreading of overstimulation of a variety of sensory receptors, development of acquired food intolerance and intolerances towards non-related allergens, e.g., to dandruff of domestic animals. Intolerance towards scents of many chemicals and furthermore towards electromagnetic field or even towards daily light may develop too but only when the disease goes chronic.

Local inflammation induced by neuropeptides will increase the expression of mRNA of transmembrane receptor molecules, e.g., transient receptor potential vanilloid 1 (TRPV1). After prolonged stimulation, the inflamed tissue becomes hypersensitive to various chemically unrelated ligands of TRPV1, as has been demonstrated by the topical application of formaldehyde to the skin of the mouse ear [25]. Probably, the overstimulation of receptors by biologicals or chemicals may lead not only to the increase the number but also to the augmented affinity of the TRPV1 ligands. From a pathophysiological standpoint, neurogenic inflammation may also be described as a small fiber neuropathy.

Small fiber neuropathy is too broad a definition that comprises many different unrelated conditions, such as diabetes mellitus, hepatitis C, sarcoidosis, paraneoplastic conditions and the exposure to neurotoxins [26]. It can be viewed that SFN is related to SBS/DMHS when the disease goes chronic. This can be manifested as MCS and hyperactivity of the airways (RADS/RUDS), or dysautonomia such as paroxysmal tachycardia, frequent urination and irritable bowel syndrome, among others, reviewed by [21]. The crossover of immune and non-immune inflammation is shown in Figure 2.

## 3. Sick Building Syndrome/Dampness and Mold Hypersensitivity Syndrome are Related to other Chronic Illnesses and Autoimmunity in Particular

Exposure to toxins may cause a great number of seemingly non-related disease manifestations, far more beyond those already accepted as respiratory problems. The spectrum of the diseases related to SBS and DMHS have been extensively reviewed [27,28,29,30,31]. These descriptions were based on a combined clinical experience of treating physicians who remained consistent in their views independently from career benefits or official acceptance.

Exposure to the secondary metabolite products of indoor air molds and mycotoxins in particular may alter the functions of not only the central and the peripheral nervous system but also the glands of internal secretion, especially the thyroid. For instance, when rats were exposed *per os* to ochratoxin A (OTA A), the blood levels of triiodothyronine (T3), thyroxine (T4) and cortisol became reduced [32,33]. Exposure to the metabolite products of indoor air molds can cause the development of autoimmune thyroidal disease and type I diabetes mellitus [34]. There are some, although scarce, reports that DMHS is related to non-thyroid illness [35]. Dennis et al. [36] published that patients exposed to indoor air molds had low T3 and/or T4 values and a low value of adrenocorticotrophic hormone (ACTH).

Some evidence that DM can cause autoimmunity of the nervous system has been published. In one study, 91 out of 119 (83%) patients presented with a peripheral neuropathy (numbness, tingling, tremors and muscle weakness) showing significantly higher titers of isotype antibodies (IgA, IgG and IgM) to neural antigens. These autoantibodies recognized myelin basic protein, myelin-associated glycoprotein (MAG), ganglioside GM1, sulfatide, myelin, oligodendrocyte glycoprotein, alfa-B-crystallin, chondroitin sulfate, tubulin and neurofilament [37]. The same group demonstrated that mixed mold mycotoxicosis has been implicated in the production of antinuclear autoantibodies (ANA) and anti-myelin antibodies against the nervous system, and autoantibodies against smooth muscles (ASM) [38]. Furthermore, chronic inflammatory demyelinating polyneuropathy (CIDP), which is an acquired immune-mediated inflammatory condition, has been diagnosed in occupants of mold-infested buildings [39]. DMHS may present with other neurological symptoms such as “brain fog”, tremors, jerking movements, spastic dysphonia, tic-like motions and idiopathic paroxysmal involuntary movements. These symptoms may be often misinterpreted or overlooked, especially in cases when conventional anti-Parkinson’s medication does not help [40]. Involvement of the nervous system is confirmed not only by detecting biochemical markers, but also by using modern functional imaging techniques [41]. At present, these novel techniques are not yet routinely implemented into clinical practice, however, I envision that this is the right direction to take.

Many indoor air mycotoxins are neurotoxic (reviewed by [30]). They have multiple actions on the cell: they may increase the production of reactive oxygen species (ROS), may deplete ATP synthesis, may alter mitochondrial membrane potential (ΔΨ) and facilitate the release of mitochondrial proteins into the cytosol. Exposure to mycotoxins activates inflammasome machinery in the cell. This will promote the production of the pro-inflammatory IL-1β cytokine, the marker of the activation of innate immunity. When IL-1β cytokine binds to its cognate receptor, IL-1R1, intracellular signal transduction will follow. This mechanism will lead to a vicious cycle of inflammation. Neuroinflammation will clinically manifest as, e.g., epilepsy [42], seizures [43] and thermoregulation problems [44], the conditions sometimes observed in DMHS.

So far, the studies on autoimmunity in SBS/DMHS are still limited; therefore, we have to collect the evidence by crumbles. We can also extrapolate some knowledge coming from studies on related conditions, but for the sake of honesty, we cannot yet claim that we have gained an indisputable proof and we do not yet know the degree of prevalence of autoimmunity in DMHS. Firstly, several reports showing the presence of some autoantibodies in relation to peripheral neuropathy were based on a small sample size: some studies were case reports. Secondly, the detection of autoantibodies is an expensive investigation that is usually not covered by public insurances and that many private DMHS patients cannot afford. Thirdly, the treating physician should already “foresee” what kind of antibodies should be looked for: we cannot take too many analyses. Now, the absence of a robust, affordable, chip-based technology to study, simultaneously, autoantibodies to multiple protein structures hampers our clinical work.

As said, we pull the evidence from crumbles and make extrapolations. In a recent large review [45], the presence of autoantibodies in fibromyalgia and CFS was combined from many relevant publications. It is to be emphasized that from the standpoint of a treating physician, both conditions associate with DMHS, and especially CFS seems quite prevalent [46]. The autoantibodies that were reviewed by Ryabkova et al. [45] were towards 5-hyrdoxytryptamine, gangliosides, phospholipids, ASM and thyroperoxidase (TPO). Further, autoantibodies towards G protein coupled receptors, against M1, M3 and M4 acetylcholine receptors (AChR) and β2-adrenergic receptors (AdR) were reported in CFS patients (reviewed by [45]). However, we should interpret these results with caution because the etiology of CFS is multivariate.

To the best of my knowledge, there is no direct indication that DMHS is linked to small fiber neuropathy (SFN), although this pathology can be suspected as the leading cause of MCS. A definite claim is impossible because skin and/or mucosa biopsies are not usually done for this group of patients. In one recent review [47], SNF was associated with vitamin deficiency, gluten sensitivity and morbidity due to toxic agents. The author reported that autoantibodies to trisulfated heparin disaccharide and fibroblast growth factor receptor 3 were found in 20% of SFN patients. Noteworthy, vitamin deficiency, gluten intolerance and exposure to biotoxins are readily diagnosed or reported by DMHS patients.

## 4. Postural Tachycardia Syndrome (POTS), Myalgic Encephalomyelitis/Chronic Fatigue Syndrome (ME/CFS) and Autoantibodies: How They are Linked?

Postural tachycardia syndrome (POTS) is a debilitating cardiovascular disorder that is characterized by an exaggerated drop of blood pressure and a heart rate increase during orthostasis. Myalgic encephalomyelitis or chronic fatigue syndrome (CFS/ME) is a severely disabling disease with no approved treatment nor social security. Sometimes, the infectious onset of CFS can be documented. Patients with ME/CFS may suffer from severe central and muscle fatigue, chronic pain, cognitive impairment and immune and autonomic nervous system dysregulation. Complex regional pain syndrome (CRPS) is a condition characterized by prolonged or excessive pain and changes in skin color and temperature, and/or swelling in the affected area. It is believed that various cytokines and autoantibodies are generated in response to stress or trauma [48].

Both POTS and ME/CFS were found common in children and youngsters exposed to indoor air DM at schools and/or at homes, occurring in 49 out of 81 (60.4%) and 56 out of 81 (69%) of respondents, respectively. By contrast, CRPS was infrequent in the reported study cohort, with a prevalence of 3 out of 81 respondents (3.7%). All the three syndromes POTS, ME/CFS and CRPS overlapped in different combinations and developed shortly after vaccination in a cohort exposed to a moldy environment [46]. In this cohort, the aetiology of POTS or ME/CFS was searched only in a few cases. Extrapolation from the current literature survey [49,50,51,52] supported the findings in these few cases. It can be therefore assumed that autoantibodies against α1-adrenergic (α1AR) and β1/2-adrenergic (β1/2AR) or acetylcholine receptors (AChR) may play a role in POTS, ME/CFS and CRPS patients who had been exposed to DM [46].

In another study, in 55 patients diagnosed with POTS, elevated serum levels of autoantibodies against the adrenergic α1 receptor and against the muscarinic acetylcholine M4 receptor were found in 89% and 53% of cases, respectively. In this study, four patients had autoantibodies against all nine receptor subtypes. A weak correlation between the clinical severity of POTS and the levels of antibodies was reported [50]. Recently, in a subset of CFS/ME patients, elevated ß2 adrenergic receptor (ß2AdR) and M3 acetylcholine receptor antibodies were documented too [51].

A strong relationship between adrenergic autoantibodies and POTS was documented in a study by Fedorowsky et al. [53]. Immunoglobulin G (IgG) derived from 17 POTS patients, 7 serum samples from patients with recurrent vasovagal syncope and 11 normal controls was analyzed for its ability to modulate activity and ligand responsiveness of α1AR and β1/2AR in transfected cells and in vivo to alter the contractility of isolated rat cremaster arterioles. In cell-based assays, IgG activation of α1AR and β1/2AR was significantly higher in POTS compared with other groups. Eight, 11 and 12 of the 17 POTS patients possessed autoantibodies that activated α1AR, β1AR and β2AR, respectively. Pharmacological blockade suppressed the IgG-induced activation of α1AR and β1/2AR. POTS IgG contracted rat cremaster arterioles, which was reversed by α1AR blockade.

Neuronal autoantibodies are not just silent bystanders but may have stimulating or inhibiting functions [53]. These autoantibodies may work, e.g., as agonists for α1-sympathetic receptors [54]. Depending on their functional properties (agonistic or antagonistic), uncontrolled vasoconstriction or vasodilatation and relaxation or spasm of smooth muscles may occur leading to a great variety of clinical manifestations. Poor adaptability of the vascular tone may lead to hypoxemia that may aggravate already existing tissue hypoxia induced by oxidative stress and energy deficit caused by mycotoxins. Hypoperfusion and hypoxia, especially of the brain, may lead to “brain fog”, cognitive impairment, central and muscular fatigue and other neurological symptoms, often reported by occupants of moldy environments [55].

## 5. Is the Detection of Mold-Specific Antibodies Useful in Diagnostics?

The immunological effects of the components of DM have been reviewed [27]. These effects are multilevel and partly controversial. Exposure to DM may either inhibit or stimulate the innate and/or adaptive arms of immunity [38,56,57,58]. The net effect depends on the composition of mycotoxins, on the growing species in the building and on the host. In different buildings, the ecological system is different and changes with time.

Many toxins are cytotoxic and may interfere with the opsonization of xenobiotics and inhibit phagocytosis by macrophages [59], making them less efficient as, e.g., in infections. The presentation of antigens by dendritic cells to T lymphocytes could be defective. For example, *Aspergillus fumigatus* inhibits the action of dendritic cells [60], the most powerful antigen presenting cells. Further, significant changes have been observed in B cell profiles [38]. Therefore, the principles of serology developed to diagnose infections are generally not well-suited to diagnose mold-related disease.

The diagnostic value of mold-specific IgG-class antibodies is low because the levels of antibody poorly correlate with the occurrence or severity of the disease. These IgG-class antibodies may indicate previous contacts only and may have some value for epidemiological studies. Type I (immediate reaction, e.g., IgE-mediated asthma, allergic rhinitis, Gell and Coombs classification) occurs less often. Atopic individuals are more prone to develop asthma through the IgE-mediated response, but only a small proportion of patients have an allergy to molds. In one study, IgE-class antibodies to different species were found in 30% of asthmatics [61]. In another study, IgE antibody positivity was reported in only 10% [62]. Thus, pulmonary problems reported by patients exposed to DM may be mediated either by allergens and allergen-specific IgE, or caused through allergen-independent mechanisms [63,64].

## 6. Why the Problems Due to the Exposure to Moisture-Damaged Buildings Persist?

This review does not focus on the discussion of controversies on what is the best way to study indoor air quality. The interested readers are referred to articles published in special journals for toxicology, environmental and occupational medicine and public health. Here, I would like to emphasize a few crucial moments that may have an impact on our interpretation and risk assessments. Correct sampling, e.g., in clinical microbiology when culturing microbes from wounds, pus or infected foreign objects in the body is crucial. *Lege artis* sampling in the environmental microbiology and toxicology is also of utmost importance. It seems however, that we are still far from optimal approaches to test indoor air.

As reported, air samples were collected from settled dust on cardboard boxes or by electrostatic dust fall collectors [65]. These techniques completely ignore the fact that toxigenic indoor microbes, such as *Stachybotrys*, emit toxins as liquid vesicles (micro vesicles, exoms) in which the concentrations of toxins are more than 1000-fold higher than the emissions coming from the same microbial particle (spores, hyphae fragments) [66]. *Trichoderma* spp. emit biotoxins, collectively called peptaibols, from water-damaged surfaces in guttation droplets in a building with indoor air problems [67]. Obviously, the concentrations measured from the dust particles may be underestimated by more than 1000-fold when compared with the real amount of toxins released by indoor water vapor. The molecular weight of the toxins may vary from 300 to 1000 Da. Mycotoxins are fat-soluble molecules and adhere to indoor micro-cavities and surfaces at low indoor air humidity. In schools or office buildings during cold seasons, e.g., at nights or weekends, the low indoor air humidity, 6–20%, does not allow the aerosolization of toxins. By contrast, when the environment is crowded in the daytime with people exhaling water vapor, the humidity rises to 60%. In this way, the occupants become exposed to mycotoxins more readily that could be estimated by studying the dust fall collectors (communication from Prof. Mirja Salkinoja-Salonen).

New methods to collect air samples that reflect better the environmental toxicity have been described [68,69]. Based on these data, it seems imperative not to use only the results of microbial culture analysis because the colonies may grow slowly and due to toxin production, some species may inhibit the growth of others. Rather, bioassay methods that estimate indoor air biotoxicity should be implemented. Such bioassays use either macrophages as the reporting tissue culture object or the motility of boar spermatozoids or other ex vivo models [60]. Some emerging reports from Finland could already associate the symptoms experienced by occupants of problematic buildings with the indoor air toxicity [55,70,71].

It is also important to collect relevant samples from humans living or working in a moldy environment. The methods studying oxidative and nitrosative stress reactions as well as the detection of biotoxins in body fluids such as urine should be made available, affordable and properly validated. Ideally, the demonstration of the same biotoxins in body fluids and in the condensed water samples collected from a problematic building will finally challenge the claim of the lack of causality. The condensed water can be collected using special air collection devices. Although this approach seems logical and technically feasible, there is a large resistance towards the development of the above-mentioned techniques. Cynically speaking, too fast a construction of inferior-quality buildings or poor building maintenance that make occupants sick and blaming them for their “wrong attitudes or exaggerated worries”, seems profitable.

## 7. Conclusions

Successful holistic treatment of patients diagnosed with DMHS or SBS as a “starting point“ for DMHS requires revolutionary re-thinking. To achieve ambitious treatment results, the medical community should abandon the current practices of putting nosologies into cages of ICD codes that are useful only for statistical purposes. Rather, DMHS should be viewed as a broad syndrome comprising different concurrent and overlapping pathophysiologies, namely the combination of autoimmunity and neuro-endocrine axis misbalance, chronic insidious inflammation, where the innate immunity orchestrates the symphony of cellular, tissue and organ dysfunctions. These dysfunctions may become irreversible. Skin and mucosa biopsies and chip-based autoantibody studies would be of utmost clinical relevance in the future.

We certainly need to focus also on the development of relevant, robust and reliable toxicological tests. The success of combating DMHS relies on transparency based on the research motivated by pure academic values without direct or indirect interferences from insurance companies or their satellites. DMHS should be viewed also as a biotoxicosis that has nothing to do with initial mental instability. DMHS is a man-made disease, therefore the goodwill of the mankind is required to build and maintain safe constructions for living and working in.

## Figures and Tables

**Figure 1 antibodies-09-00026-f001:**
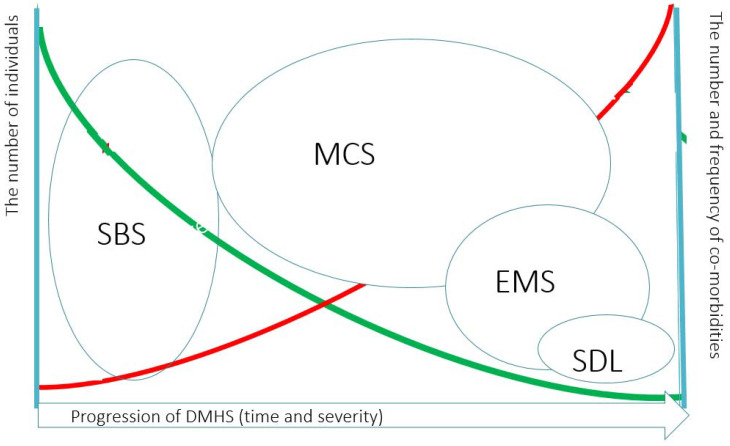
The clinician point of view of the pyramid of the progression of Dampness and Mold Hypersensitivity Syndrome and how it is related to Sick Building Syndrome. Dampness and Mold Hypersensitivity Syndrome (DMHS) progresses with prolonged or cumulative exposure to indoor air dampness microbiota. Simultaneous exposure to other xenobiotics may foster the development of symptoms. At some time point SBS becomes irreversible and at this stage the patient may develop Multiple Chemical Sensitivity (MCS). Some fraction of patients will develop in addition to MCS the Electromagnetic Sensitivity (EMS) and a very small fraction of patients have sensitivity to daily light (SDL). Noteworthy, MCS and EMS may overlap. The green line indicates the relative prevalence of severe hypersensitivity manifestations in DMHS. The red line indicates the relative frequency and the numbers of concomitant co-morbidities with the development of severe hypersensitivity.

**Figure 2 antibodies-09-00026-f002:**
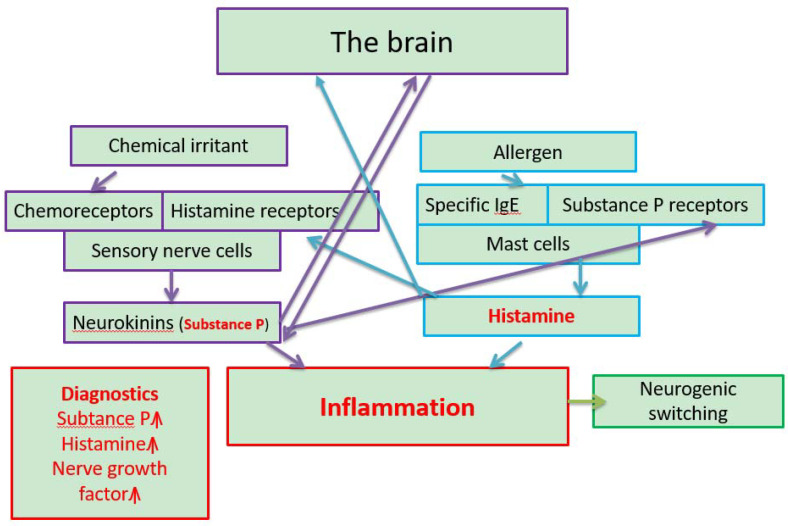
The interplay between allergic (boxes with blue lines) (IgE-mediated) and non-allergic neurogenic inflammation (boxes with violet lines) in Multiple Chemical Sensitivity (MCS) related to the Dampness and Mold Hypersensitivity Syndrome (DMHS). Customized from” The role of neurogenic inflammation in chemical sensitivity”, Meggs, W.J. *Ecopsychology*
**2017**, *9*, 83–89. MCS is the 4th criterion of the DMHS (Valtonen, Front Immunol **2017**). When a person is exposed to xenobiotics that are generally small molecules (hundreds of Da), those bind to chemoreceptors that may be hyper-activated due to previous prolonged or cumulative exposures to environmental toxins. This binding results in the release of potent messengers of neurogenic inflammation, the so-called neurokinins, e.g., Substance P (SP). SP may pass through the blood brain barrier and induce neuroinflammation in the brain. Also, SP binds to SP receptors on mast cells that upon activation release histamine. When a person encounters protein from dampness microbiota (thousands of Da) to which (s) he might have been already sensitized (the presence of IgE-antibodies) mast cells become activated through the binding of IgE-antibody complexes to e.g FcεR1 receptors. The release of histamine causes systemic inflammation and augments the hyper-activation of sensory receptors. Histamine and inflammation are central to the pathology of MCS (highlighted in red). Histamine may pass directly to the brain. Neurogenic switching can occur as the inflammation may proceed to organs (e.g., skin or g-i tract) other than the cite of inoculation (e.g., airways) (Meggs, **2017**). The proposal for biomarkers is presented in a separate box with red lines.

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
