# Peer review of "The Roles of Autoimmunity and Biotoxicosis in Sick Building Syndrome as a “Starting Point” for Irreversible Dampness and Mold Hypersensitivity Syndrome"

_2073-4468, 2020, doi:10.3390/antib9020026_

Round 1
Reviewer 1 Report
Sick Building Syndrome (SBS) characterized with various nonspecific symptoms that occur in the occupants of certain buildings. This review performed nice literature researching with the aim to bridge SBS with the exposure of indoor air dampness microbiota. The progress of SBS may lead to irreversible Dampness and Mold Hypersensitivity Syndrome (DMHS). Based on that, the author proposed the roles of autoimmunity in transition between SBS and DMHS. This is well-prepared review with lot of thoughtful conception to improve our understanding of SBS and DMHS. I predict it will be useful for clinical researchers and scientists who may interest in SBS and DMHS.
Author Response
I found no critics, nor correction suggestions from the Reviewer1. I want to express my gratitude for reading it and giving me a feedback
Reviewer 2 Report
The paper is devoted to the controversial topic of “Sick Building Syndrome” associated with Autoimmunity and Biotoxicosis.
The abstract is presented as for original article. The last sentence in the abstract is confusing.
The title is informative and relevant, as well as the references are relevant and recent. The cited sources are referenced correctly. Appropriate and key studies are included.
However, there are many speculations in this review article, especially the part for autoimmunity and autoantibodies related to autoimmune encephalitis.
The conclusions are supported by references and own decisions.
Author Response
Thank you reviewer 2 for reading my review paper. You are right there are speculations. In science we have to do speculations and to test our hypothesis. at present we collect clinical material to test our hypothesis.
I will correct the last sentence in the abstact
Round 2
Reviewer 2 Report
Dear authors,
I think that the paper is not suitable for the journal, and the topic is controversial, the input of autoimmunity is not appropriate.
Not all requirements are accomplished.
Author Response
Reviewer 1 report 1:
The paper is devoted to the controversial topic of “Sick Building Syndrome” associated with Autoimmunity and Biotoxicosis.
The abstract is presented as for original article. The last sentence in the abstract is confusing.
I have changed the sentence into “Association of autoimmunity with DMHS should be confirmed through cohort studies preferably using chip-based technology”.
The title is informative and relevant, as well as the references are relevant and recent. The cited sources are referenced correctly. Appropriate and key studies are included.
However, there are many speculations in this review article, especially the part for autoimmunity and autoantibodies related to autoimmune encephalitis.
The conclusions are supported by references and own decisions.
Reviewer 1 report 2:
Dear authors,
I think that the paper is not suitable for the journal, and the topic is controversial, the input of autoimmunity is not appropriate.
Not all requirements are accomplished.
I do not see any other requirements. I present the paper that I have been working on for a long time. The Editor asked me to present a paper. I extended the subject on autoantibodies. Although this subject may seem controversial, I wanted to raise alert that this topic warrants more studies. I also discuss the techniques that need to be improved.
Reviewer 2 report 1:
Sick Building Syndrome (SBS) characterized with various nonspecific symptoms that occur in the occupants of certain buildings. This review performed nice literature researching with the aim to bridge SBS with the exposure of indoor air dampness microbiota. The progress of SBS may lead to irreversible Dampness and Mold Hypersensitivity Syndrome (DMHS). Based on that, the author proposed the roles of autoimmunity in transition between SBS and DMHS. This is well-prepared review with lot of thoughtful conception to improve our understanding of SBS and DMHS. I predict it will be useful for clinical researchers and scientists who may interest in SBS and DMHS.
Academic editor report:
The content of the review article "The Roles of Autoimmunity and Biotoxicosis in Sick Building Syndrome as a “starting point” for irreversible Dampness and Mold Hypersensitivity Syndrome" is very interesting and new. However, there are some parts that can be fixed to make the thread of writing more fluid.
The aim is: Describe SBS in connection with the exposure to indoor air dampness microbiota (DM). However, the logical sequence is not aways maintained, whereas the review should be more focused on the aim.
I had to extend my review into the topic of the role of autoantibodies to make the review suitable for the journal The meaning of Sick Building Syndrome is decribed in rows 56-81.
The exposition is sometimes a bit complicated and confused.
Sentences from lanes 141-151 are not really clear.
I rerwote this paragraph. In red you may see the changes.
Exposure to the secondary metabolite products of indoor air molds and mycotoxins in particular may alter the functions of not only the central and the peripheral nervous system but also the glands of internal secretion, especially the thyroid. For instance, when rats were exposed per os to ochratoxin A (OTA A) the blood levels of triiodothyronine (T3), thyroxine (T4) and cortisol became reduced [32,33]. The exposure to metabolite products of indoor air molds can cause the development of autoimmune thyroidal disease and type I diabetes mellitus [34]. There are some, although scarce reports, that DMHS is related to Non-Thyroid Illness [35]. Dennis et al. [36] have published that patients exposed to indoor air molds had low T3 and/or T4 values and low value of adrenocorticotrophic hormone (ACTH).
Sentences lanes 173-175 seem not well connected wth the rest.
I attempted to make this part clearer. The changes are in red
Many indoor air mycotoxins are neurotoxic [reviewed 30]. They have multiple actions on the cell: they may increase the production of reactive oxygen species (ROS), may deplete ATP synthesis, may alter mitochondrial membrane potential (ΔΨ), and facilitate the release of mitochondrial proteins into the cytosol. Exposure to mycotoxins activates inflammasome machinery in the cell. This will promote the production of the pro-inflammatory IL-1β cytokine, the marker of the activation of innate immunity. When IL-1β cytokine binds to its cognate receptor, the IL-1R1, intracellular signal transduction will follow. This mechanism will lead to a vicious cycle of inflammation. Neuroinflammation will clinically manifest as e.g. epilepsy [42], seizures [43] and thermoregulation problems [44], the conditions sometimes observed in DMHS.
Paragraph: Postural tachycardia syndrome (POTS), Myalgic Encephalomyelitis/Chronic Fatigue Syndrome 273 (ME/CFS) and autoantibodies: how they are linked? " seem not well connected with the rest of the review. It can be shortened and made more clear?
I shortened this paragraph. The mention of these three syndromes is very important because they are linked to irreversible DMHS. It is also important to mention the role of neuronal antibodies. Unfortunately, this investigation is at the very beginning. At present, we are collecting sera to confirm this finding.
The changes are in red.
Both POTS and ME/CFS were found common in children and youngsters exposed to indoor air DM at schools and/or at homes occurring in 49 out of 81, 60,4% and 56 out of 81, 69% of respondents, respectively. By contrast, CRPS was infrequent in the reported study cohort, with a prevalence of three out of 81 respondents (3.7%). All the three syndromes POTS, ME/CFS and CRPS overlapped in different combinations and developed shortly after vaccination in a cohort exposed to mouldy environment [46]. In this cohort the aetiology of POTS or ME/CFS was searched only in a few cases. Extrapolation from the current literature survey [49-52] supported the findings in these few cases. It can be therefore assumed that autoantibodies against α1-adrenergic (α1AR) and β1/2-adrenergic (β1/2AR) or acetylcholine receptors (AChR) may play a role in POTS, ME/CFS and CRPS patients who had been exposed to DM [46].
Neuronal autoantibodies are not just silent bystanders but may have stimulating or inhibiting functions [53]. These autoantibodies may work e.g. as agonists for α1-sympathetic receptors [54].
I thank the reviews and the Editor for their comments. I hope the review has been improved and can be accepted for the publication..
Sincerely
Tamara Tuuminen.
P. S. unfortunately the program changed the red colour whic hI used to indicate changes
